# Genomic epidemiology of enteropathogenic *Escherichia coli* in southwestern Nigeria

**Olabisi C. Akinlabi**[1¤a], **Rotimi A. Dada**[1,2], **Ademola A. Olayinka**[3], **Ibukunoluwa O. Oginni-Falajiki**[3], **Oyeniyi S. Bejide**[1¤b], **Pelumi D. Adewole**[1¤c], **Nicholas R. Thomson**[4,5], **Aaron O. Aboderin**[3,6], **Iruka N. Okeke**[1]*

1 Department of Pharmaceutical Microbiology, Faculty of Pharmacy, University of Ibadan, Ibadan, Nigeria,
2 Department of Pharmaceutical Microbiology and Biotechnology, Ahmadu Bello University, Zaria, Nigeria,
3 Department of Medical Microbiology and Parasitology, Obafemi Awolowo University, Ile-Ife, Nigeria,
4 London School of Hygiene and Tropical Medicine, London, United Kingdom, 5 Parasites & Microbes Programme, Wellcome Sanger Institute, Wellcome Genome Campus, Cambridge, United Kingdom,
6 Department of Medical Microbiology and Parasitology, Obafemi Awolowo University Teaching Hospitals Complex, Ile-Ife, Nigeria

¤a Current Address: Department of Medical Microbiology and Infectious Diseases, University of Manitoba, Winnipeg, Canada
¤b Current Address: Augustine University, Ilara Epe, Lagos, Nigeria.
¤c Department of Medical Laboratory Science, College of Health Sciences, Bowen University, Iwo, Osun State, Nigeria
* iruka.n.okeke@gmail.com

## Abstract

### Background

Enteropathogenic *Escherichia coli* (EPEC) are etiological agents of diarrhea. We studied the genetic diversity and virulence factors of EPEC in southwestern Nigeria, where this pathotype is rarely characterized.

### Methodology/Principal findings

EPEC isolates (n = 96) recovered from recent southwestern Nigeria diarrhea case-control studies were whole genome-sequenced using Illumina technology. Genomes were assembled using SPAdes and quality was evaluated using QUAST. Virulencefinder, Ectyper, and ResFinder were used to identify virulence genes, serotypes, and resistance genes. Multilocus sequence typing was done by STtyping. Single nucleotide polymorphisms (SNPs) were called out of whole genome alignment using SNP-sites and a phylogenetic tree was constructed using IQtree. Thirty-nine of the 96(40.6%) EPEC isolates were from diarrhea cases diarrhea. Nine isolates from diarrhea patients and four from healthy controls were typical EPEC, harboring bundle-forming pilus (*bfp*) genes whilst the rest were atypical EPEC. There were 15 EPEC-EAEC hybrids. Atypical serotypes O71:H19 (16, 16.6%), O108:H21 (6, 6.3%), O157:H39 (5, 5.2%), and O165:H9 (4, 4.2%) were the most prevalent; only 8 (8.3%) isolates belonged to classical EPEC serovars. The largest, ST517 clade harbored

**Data availability statement:** Sequence data were submitted to ENA and are available as Bioproject PRJEB8667 at ENA https://www.ebi.ac.uk/ena/browser/view/PRJEB8667 and Genbank https://www.ncbi.nlm.nih.gov/bioproject/PRJEB8667/. Tree and metadata are available in interactable format at https://microreact.org/project/hnPuLKV1d7pnskcG-146MJj-epec-isolates-and-their-locations-2025.

**Funding:** This research was supported by African Research Leader award MR/L00464X/1 to INO and NRT from the UK Medical Research Council (MRC) and the UK Department for International Development (DFID) under the MRC/DFID 23 concordat agreement that is also part of the EDCTP2 programme supported by the European Union https://gtr.ukri.org/projects?ref=MR%2FL00464X%2F1. INO is a Calestous Juma Science Leadership fellow supported by the Gates Foundation (INV-036234) https://gcgh.grandchallenges.org/grant/leveraging-bacterial-genomics-health-solutions-africa. The funders had no role in study design, data collection and analysis, decision to publish, or preparation of the manuscript.

**Competing interests:** The authors have declared that no competing interests exist

multiple siderophore and serine protease autotransporter genes and included an O71:H19 subclade <10 SNPs apart, representing a likely outbreak involving 15 children, four with diarrhea. Likely outbreaks, of typical O119:H6(ST28) and atypical O127:H29(ST7798) were additionally identified.

## Conclusion/Significance

EPEC circulating in southwestern Nigeria are diverse and differ substantially from well-characterized lineages seen previously elsewhere. EPEC carriage and outbreaks could be commonplace but are largely undetected, hence, unreported, and require genomic surveillance for identification.

### Author summary

Enteropathogenic *Escherichia coli* (EPEC) are causes of diarrhea, particularly in infants. We sequenced and analyzed the genomes of EPEC from recent epidemiological studies in southwestern Nigeria. Majority of the isolates lacked the bundle-forming pilus genes that are used to categorize EPEC as typical and belonged to subtypes rarely or never reported in the literature. Our data show that three unrecognized outbreaks occurred in the course of the epidemiological surveys and that the largest one was caused by a poorly defined lineage. Genomic methods are needed to fully understand the epidemiology of EPEC Nigeria and nearby African countries.

## Introduction

Diarrheagenic *Escherichia coli* (DEC), are prominent causes of diarrhea in children under five years of age with enterotoxigenic *E. coli* (ETEC), enteroaggregative *E. coli* (EAEC) and enteroinvasive *E. coli* (EIEC) pathotypes being the most common [1]. Enteropathogenic *E. coli* (EPEC), one of the first defined diarrhoeagenic pathotypes, has largely been eliminated as a significant public health threat in North America and Europe but remains a common cause of infantile diarrhea in Africa, Asia, and South America [2–10]. The EPEC pathotype was originally defined in the context of diarrhea outbreaks among young children [11]. Other DEC pathotypes, including, enterohemorrhagic *E. coli* (EHEC) and Shiga toxin-producing *E. coli* (STEC), enterotoxigenic *E. coli* (ETEC), enteroinvasive *E. coli* (EIEC) and diffusely adherent *E. coli* (DAEC), are also differentiated based on their virulence factors, mechanism of disease, and clinical features [12]. EPEC is infrequently reported from Africa [9,13,14] largely because of the difficulty in delineating EPEC from commensals. Earlier studies identified EPEC by serotyping [15–17]. However, this method does not capture EPEC that belong to serotypes that rarely or never circulated in the Western hemisphere since EPEC-identifying serotyping sets focus on only these so-called classical serovarieties [18–20]. Serovarieties represent clades descended from a common ancestor

that produce similar O (somatic) and H (flagella) antigens (EPEC typically lack K antigens) [15]. So-called Classical EPEC majorly belong to O serogroups O55, O86, O111, O119, O125, O126, O127, O128, and O142 [16,18,21]. However, within these serogroups, only certain serotypes are actually EPEC, for example, O55:H6, O86:H34, O111:H2, O119:H2, O119:H6, O125:H6, O126:H2, O127:H⁻, O127:H6, O128:H2, and O142:H6 [18]. The O51:H ⁻, O114:H2, O127:H40 and O142:H34 serotypes are more recently discovered typical EPEC that are often considered 'classical' as well [17]. Identifying EPEC by their ability to produce attaching-and-effacing lesions on cultured human epithelial cells (the HEp-2 cell line) represents a reliable but time- and resource-intensive means to identify this pathogen and has commonly been employed in South and Central America [22]. Attaching-and-effacing lesions are produced by a collection of virulence factors, including intimin encoded by the *eae* gene on the Locus for Enterocyte Effacement (LEE) pathogenicity island [23]. Intimin is required for intestinal colonization in the host and the EPEC-defining attaching-and-effacement phenotype. Different intimin subtypes are classified based on the variable 280-amino acid C-terminal sequence of intimin ($Int_{280}$) [24,25]. Enteropathogenic *E. coli* are classified as typical or atypical EPEC based on the presence of the EPEC adherence factor (EAF) plasmid, which carries bundle-forming pili (*bfp*) and plasmid-encoded regulator (*per*) genes [26]. The diarrheagenicity of typical EPEC (tEPEC) isolates has been confirmed in adult volunteer challenge studies [27], but less evidence is available to understand the virulence of atypical strains [28]. However, atypical EPEC (aEPEC) represent far more lineages and carry a broad range of potential virulence loci, so more research is likely to delineate additional hypervirulent lineages and virulence genes [25]. Their virulence is also alluded to by the numerous outbreak reports involving aEPEC from around the world, including Japan, the United States, and China, highlighting its public health significance beyond endemic settings [29–31]. EPEC isolates may also express other virulence factors including, but not limited to non-LEE effectors (Nle) and autotransporters such as EspC, EspI*, EspP*, Pet*, Pic and Sat [32].

Given that the genetic basis for EPEC virulence is well understood, molecular methods to identify LEE and bundle-forming pilus genes represent the most commonly used approaches for its identification, we recently reported that PCR methods developed in other parts of the world produced false-positives and negatives compared to whole genome sequencing in Nigeria [33]. Based on genetic diversity, evolutionary links, and virulence characteristics, EPEC is divided into several phylogenetic lineages, designated as EPEC1 through EPEC10. [8,34–36]. MLST and then whole genome sequence approaches have since confirmed these EPEC lineages and uncovered a total of 15 EPEC phylogenomic groups [8,34–37]. These phylogenetic classifications assist researchers in comprehending EPEC evolution, host interactions, and pathogenic potential, facilitating the formulation of focused diagnostic and intervention techniques. Here, we analyze whole-genome sequence-identified isolates from four diarrheal studies conducted in Nigeria. In this study, we characterized EPEC isolates from multiple epidemiological studies and investigated potential outbreaks, including one involving at least 19 children.

## Methodology

**Ethics Statement**: Ethical approval for this research was received from UI/UCH Ethics Committee with assigned numbers UI/EC/15/0093 and UI/EC/18/0335, and from Obafemi Awolowo University Teaching Hospital Complex with assigned number IRB/IEC/0004553. Written informed consent was obtained from parents or guardians of all children, and from adults, from whom the isolates were collected.

**Isolates analyzed**: This study analyzed the genomes of 96 EPEC isolates isolated from four different case-control studies [38]. Three of these studies (two of which are published [13,14]) were conducted in Ibadan, Oyo State, while the fourth was carried out in Ile-Ife, Osun State [34]. Three of the studies' stool specimens were collected from both children with diarrhea and healthy controls while the fourth study, involved the collection of stool specimens from people living with HIV—primarily adults—as well as HIV-negative patients with diarrhea, both groups recruited from two health care facilities in Ibadan. Isolates from children with diarrhoea were identified as EPEC by screening their whole genome sequences for *eae* and *bfp* genes using VirulenceFinder [36].

**Whole genome sequencing:** Isolates from children in Ile-Ife were whole genome sequenced using the same protocol employed for sequencing the Ibadan isolates [14]. Briefly, *E. coli* genomic DNA was isolated using the Wizard Genomic Extraction kit (Promega) according to the manufacturer's instructions. The DNA extracts were quantified using a dsDNA broad range quantification assay by Nanodrop (ThermoFisher) and transported to the Wellcome Trust Sanger Institute for sequencing on the Illumina MiSeq instrument using the 600-cycle v3 MiSeq reagent kit and the libraries were prepared using the Illumina Nextera XT DNA Library Prep Kit. The SPAdes assembler [39] was used to assemble raw data in Nigeria, and the assembly quality was assessed using the Quality Assessment Tool for Genome Assemblies (QUAST) [40] and CheckM [41]. Speciation was performed using BactInspector v0.1.3 (https://gitlab.com/antunderwood/bactinspector/).

**Whole genome sequence analyses:** All sequence analysis was carried out using the Command-line interface. Virulence factors and plasmid replicons were detected using VirulenceFinder and PlasmidFinder, respectively [42,43]. Resistance genes were identified using the ARIBA and ResFinder databases [44]. Isolates carrying any *eae* allele but lacking Shiga toxin genes were classed as EPEC. The Intimin alleles were delineated by blasting the *eae* sequences with the NCBI database and further clicking on the Genbank information to give details on the intimin type [45]. Achtman scheme 7 loci scheme Multilocus sequence typing (MLST) [46] was performed using STtyping [47]. *In silico* serotyping was carried out using Ectyper [48]. Phylogrouping was carried out using the Waters et al's (2020) CommandLine Tool Ezclermont [49]. Isolates belonging to serogroups, O55:H6, O86:H34, O111:H2, O119:H2, O119:H6, O125:H6, O126:H2, O127:H⁻, O127:H6, O128:H2, O142:H6 [18] O51:H⁻, O114:H2, O127:H40, were classed as classical EPEC and others, as non-classical [18,26]. Sequence data were submitted to ENA and are available as Bioproject PRJEB8667 at ENA https://www.ebi.ac.uk/ena/browser/home and Genbank https://www.ncbi.nlm.nih.gov/genbank/.

An EPEC phylogenetic tree was constructed by mapping 67 international reference EPEC genomes [50] which is a representatives of the known EPEC groups and our genomes with prototypical EAEC 042 strain (FN65554) using an in-house Wellcome Sanger Institute tool for mapping reference to reads - sh16_scripts (https://github.com/sanger-pathogens/sh16_scripts), SNPs were called with snp-sites [51]. Phylogenetic trees were then constructed using IQTree [52] using GTR+ASC+G model.

**Nutritional status**: Nutritional status at the time isolates were recovered was determined by computing weight for height, height for weight and weight for age from data collected with the specimens and interpreting them according to WHO criteria https://www.who.int/data/nutrition/nlis/info/malnutrition-in-children.

### Statistical analysis

For data analysis, Epi Info version 7 software (Centers for Disease Control and Prevention, Atlanta, GA, USA) was used. Fisher's exact and Chi-square tests were employed to examine the relationship between cases and controls. p-values <0.05 were considered statistically significant.

## Results

### Classification of EPEC isolates

A total of 96 EPEC isolates, listed with identified characteristics in S1 Table, were isolated from all four studies, with 93/96 (96.9%) from children and 3/96 (3.1%) from adults. Among these, 39/96 (40.6%) were from individuals with diarrhea and 57/96 (59.4%) from apparently healthy controls. EPEC were isolated from children presenting at Ibadan's University College Hospital Ibadan (UCH) 9/96 (9.4%), Primary Health Care centres in Ibadan 24/96 (25.0%), and primary, secondary, and tertiary hospitals in Ile-Ife 25/96 (26.0%) in Ile-Ife (Table 1). Adult samples were from people living with HIV (PLWHIV) attending UCH's Infectious Disease Institute 3/96 (3.1%) or from HIV-negative patient controls at other UCH clinics, as shown in Table 1. Eleven of the isolates were from children under one year of age with diarrhea and 19 from healthy children in the same age bracket. EPEC was not associated with diarrhea among study participants in any age range, including children under 1 year, children aged 0–6 months and children aged 7–12 months (Table 1).

**Table 1. Age distribution of individuals from whom EPEC were recovered.**

|  | Individuals with diarrhea (n = 975) | Controls without diarrhea (n = 932) | Total (n = 1907) | p-value* |
|---|---|---|---|---|
| 0 to 3 months | 6 | 4 | 10 | 0.806 |
| 4 to 6 months | 0 | 5 | 5 | 0.0654 |
| 7 to 9 months | 3 | 6 | 9 | 0.4616 |
| 10 to 12 months | 2 | 4 | 6 | 0.6424 |
| 13 and above | 8 | 16 | 24 | 0.0998 |
| Adult | 3 | 0 | 3 | 1.0000 |
| Unknown | 2 | 2 | 4 | ND |

*Chi-squared or Fisher's Exact test comparing recovery from individuals with diarrhea to recovery from controls without diarrhea.

More 57/96 (59.4%) of the EPEC isolates were from healthy individuals (Table 2) and the majority 83/96(3.9%), of the isolates were atypical EPEC, belonging to non-classical EPEC serotypes. However, typical EPEC, which were uncommon overall, were recovered from 9/96 (1.1%) individuals with diarrhea and 4/96 (0.3%) apparently healthy controls ($p < 0.05$, Table 3), implying that their recovery was significantly associated with diarrhea.

### Phylogroup, ST, intimin type and Serogroups of the EPEC isolates

Forty-six of 96 (47.9%) of the EPEC isolates belonged to *E. coli* phylogroup B1. Of the others 20/96 (20.8%), 17/96 (17.7%), 5/96 (5.2%), 1/96 (1%) were from phylogroups B2, A, D and E, respectively. In addition, 3/96 (3.1%) were untypeable and none belonged to phylogroups C or F (Table 3). All the identified typical EPEC belonged to phylogroups A, B1, and B2 none from D, E, and U, while all classical EPECs belonged to phylogroup B2 (Fig 1).

The EPEC isolates belonged to 26 different sequence types (STs), with ST517 being the most common. ST517 isolates were recovered from three different recruiting sites (IFE, UCH and PHC) as shown in Fig 1. Typical EPEC isolates belonged to ST28, ST206, 328, 2346 and 4128. Atypical EPEC were distributed among the seven STs illustrated in Fig 1. Classical EPEC belonged to two STs: ST28 (O119:H6) and ST2346 (O142:H34), with all ST28 EPEC were from children with diarrhea while ST2346 were all from healthy children. The classical isolates were all typical EPEC carrying both *bfp* and *eae* genes (Fig 1).

**Table 2. EPEC isolates isolated from recent southwest Nigeria epidemiological studies.**

| Source studies (Reference) | Study Type | Cases with diarrhea | Controls without diarrhea | Total | Cases with diarrhea from which EPEC were recovered | Controls Without diarrhea from which EPEC were recovered | EPEC n (%) | p-value* |
|---|---|---|---|---|---|---|---|---|
| Ibadan Primary Health Care Center (PHC) [14] | Children aged under 5 | 120 | 357 | 477 | 7 | 17 | 24(5.0%) | 0.6331 |
| University Teaching Hospital Ibadan (UCH) | Children aged under 5 | 359 | 241 | 600 | 6 | 3 | 9(1.5) | 0.9372 |
| Ife primary, secondary, and tertiary hospitals (IFE) [38] | Children aged under 5 | 167 | 334 | 501 | 8 | 17 | 25(5.0%) | 1.000 |
| Infectious Disease Institute (IDI) [13] | People living with HIV | 202 | N/A | 202 | 1 | N/A | 3(0.9%) | 0.6837 |
| Infectious Disease Institute (IDI) [13] | HIV-negative individuals | 127 | N/A | 127 | 2 | N/A |  |  |
| Total |  | 975 | 932 | 1907 | 24 | 37 | 61(3.2%) |  |

*Chi-squared or Fisher's Exact test comparing recovery from individuals with diarrhea to recovery from controls without diarrhea.

**Table 3. Identified EPEC from the studies.**

| EPEC category | Defining criteria for category | Case isolates from patients n=805 (%) | Control isolates from symptomless individuals n=1311 (%) | Total isolates from 2116 individuals (%) | p-value* |
|---|---|---|---|---|---|
| Typical | Carry the LEE and *bfp* genes | 9/805 (1.1) | 4/1311 (0.3) | 13/2116 (0.6) | **0.0417** |
| Atypical | Carry the LEE but lack *bfp* and other virulence plasmid genes | 30/805 (3.7) | 53/1311 (4.0) | 83/2116 (3.9) | 0.8040 |
| Classical | Serotyped *in silico* as O55:H6, O86:H34, O111:H2, O119:H2, O119:H6, O125:H6, O126:H2, O127:H-, O127:H6, O128:H2, O142:H6 [18] O51:H-, O114:H2, O127:H40 | 5/805 (0.6) | 3/1311 (0.2) | 8/2116 (0.4) | 0.2879 |
| Non-classical | Belong to any serotypes other than in the cell above | 34/805 (4.2) | 54/1311 (4.1) | 88/2116 (4.2) | 0.9111 |
| All EPEC | Have the LEE and lack Shiga toxin genes | 39/805 (4.8) | 57/1311 (4.3) | 96/2116 (4.5) | 0.6703 |

*Chi-squared or Fisher's Exact test comparing isolates from individuals with diarrhea isolates from controls without diarrhea.

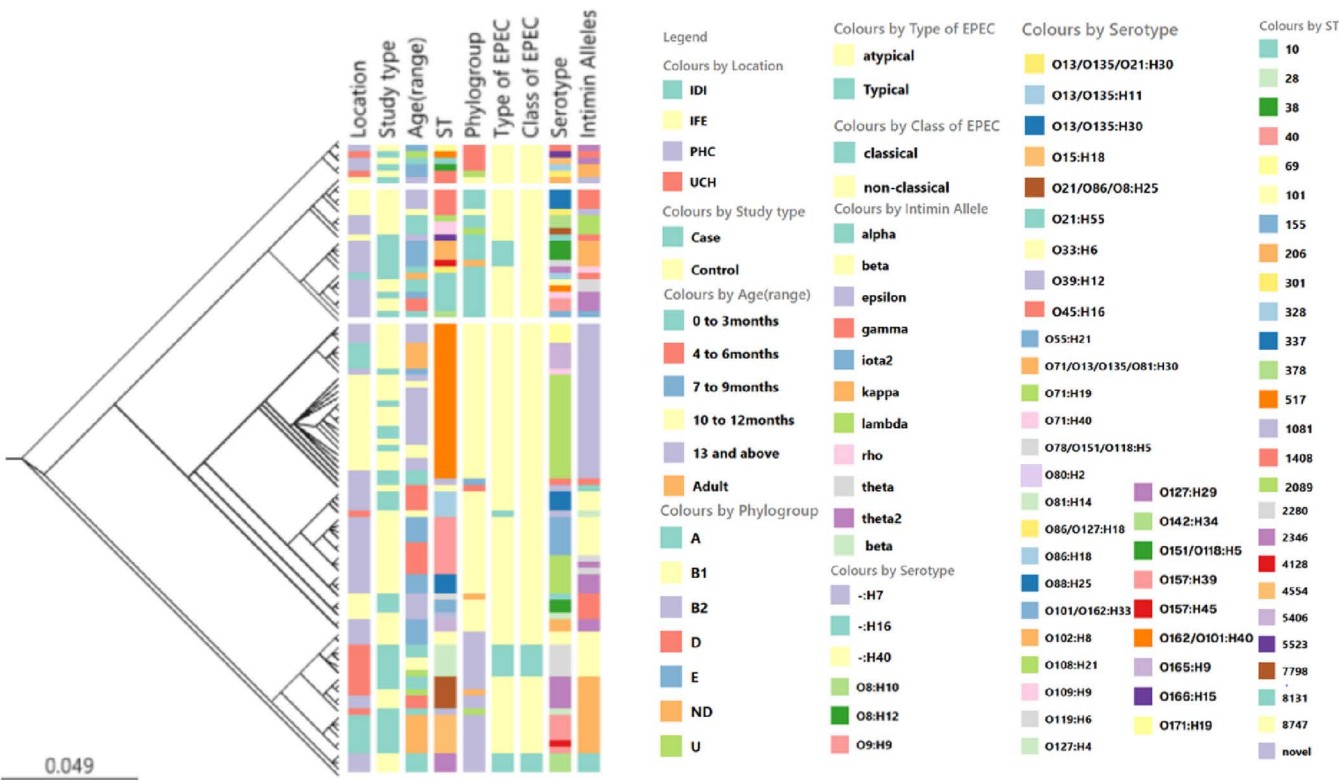

**Fig 1. Maximum-likelihood whole-genome SNP-based tree of EPEC isolates from this study juxtaposed against source (location, age and health status of patient) and classification (typical/ atypical, classical or non-classical serovar, intimin type, phylogroup and ST) criteria.** Data are available in interactable format at https://microreact.org/project/hnPuLKV1d7pnskcG146MJj-epec-isolates-and-their-locations-2025.

A total of 39 serotypes were identified, with serotype O71:H19 as the most prevalent 16 of 96 (16.7%). EPEC belonging to the O71:H19 serotype were isolated from Ile-Ife (IFE) and Ibadan Primary Health Centers (PHC) (Fig 1). The typical EPEC isolates belonged to both classical serotypes O119:H6 (n=5), O142:H34 (n=3), and non-classical O151/O118:H5 (n=4), while other serotypes were atypical EPEC.

Ten different intimin types were identified from the EPEC genomes in this study, with epsilon being the most prevalent [26 of 96 (27.1%)], followed by beta [17 of 96 (17.7%)], theta2 and gamma [11 of 96 (11.4%)] (Fig 1). All typical EPEC genomes encoded intimin beta or kappa but a much broader range of intimin alleles was seen among atypical isolates (Fig 1).

None of the isolates in the study belonged to the earliest EPEC phylogenomic groups, EPEC1 and EPEC2, originally delineated by multilocus enzyme electrophoresis Most isolates in this study belong to new phylogenomic lineages that are yet to be described in the literature, however, some belonged to EPEC4 7(7.7%), EPEC5 3(3.3%), EPEC7 4(4.4%), EPEC9 3(3.3%), and EPEC10 7(7.7%). The majority of our isolates are closely related to isolates from Gambia, belonging to an as yet unnamed lineage [37] (Fig 2).

**EPEC hybrids.** To determine if the EPEC identified in this study carried genes associated with other *E. coli* pathotypes we searched their genomes using VirulenceFinder. None of the isolated EPEC isolates carried Shiga toxin genes; no enterohaemorrhagic *E. coli* were recovered in any of the epidemiological surveys. However, most of the studies yielding the isolates were focused on infants, from whom EHEC are rarely recovered. One O101/O162:H33 isolate did carry the enterohemolysin gene, *ehx*, which is typically contained on an EHEC virulence plasmid. An ST378 phylogroup isolate is the only one bearing the iota allele of intimin [53]. The isolate was recovered from a child attending an Ibadan primary health center. Isolates belonging to this clone carry aerobactin- and yersiniabactin-encoding siderophore genes, a range

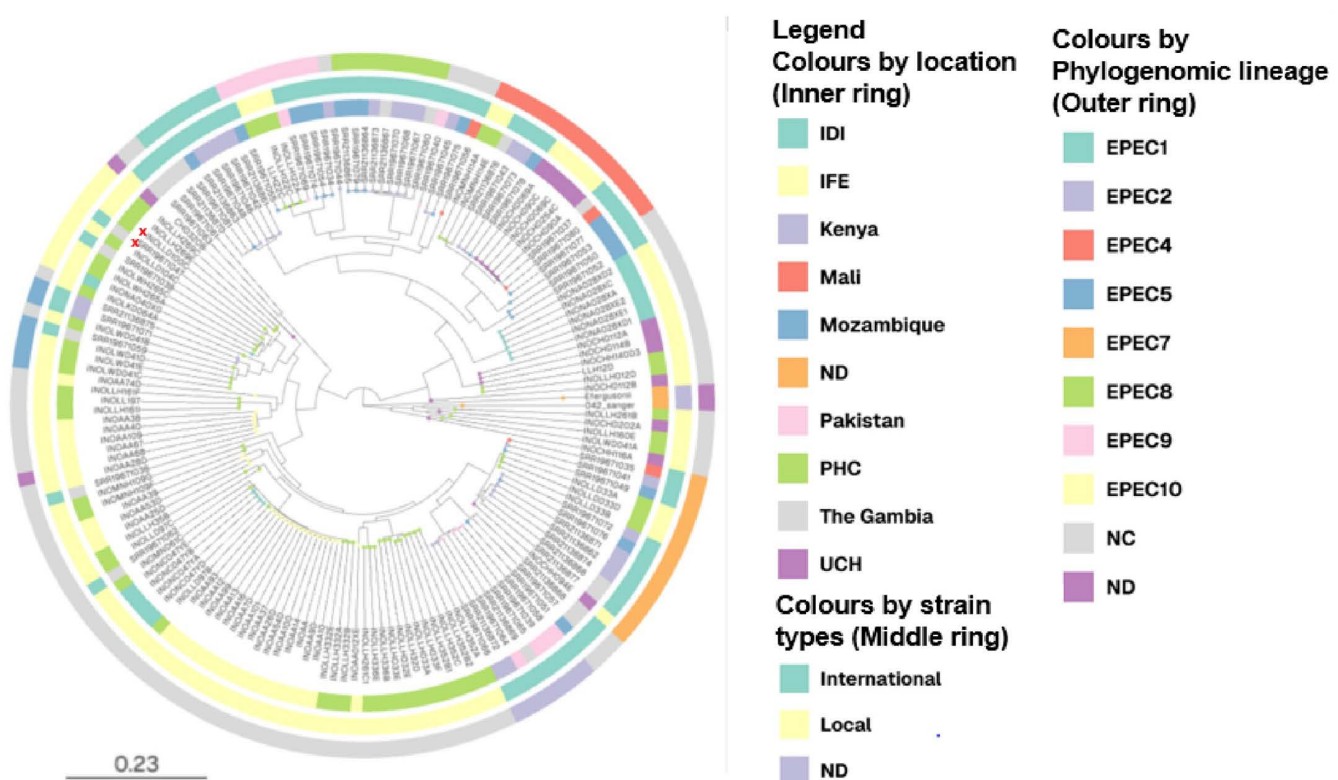

**Fig 2. Maximum-likelihood whole-genome SNP-based tree and results of phylogenomic analysis of 61 EPEC isolates from Nigeria in this study and 67 international strains.** The inner ring represents the isolation sources and location, the middle ring represents the strain types, with the locally isolated strains in yellow, outer ring represent the phylogenomic lineages, ND; not determined Local isolates, NC; not classified international strains. Isolates from the Gambia that are closely related to isolates from primary health care centres in this study are marked with red crosses. Data are available in interactive format at: https://microreact.org/project/usSgwCwVrm1AZ76kHqfk1d-epec-local-and-international-strains.

of non-LEE effector (*nle*) genes and *hlyE* but lack serine protease autotransporter of the Enterobacteriaceae (SPATE)-encoding genes [53]. Our own O101/O162:H33 isolate fit this profile and also carry enteroaggregative *E. coli* (EAEC) aggregative adherence fimbriae I (*agg*) genes and their regulator *aggR*. Thus, it is a hybrid EPEC-EAEC isolate, although it lacked other virulence genes common in EAEC such as SPATEs, the antiaggregative protein (*aap*) gene and its secretion system encoded by *aat* genes (Fig 3).

A total of 15 EPEC isolates evaluated in this study, including the O101/162:H33 isolate, co-harbored EAEC virulence genes (Fig 3) such as the EAEC hexosyltransferase homolog *(capU)* - 9(8%), enteroaggregative immunoglobulin repeat protein (*air*) - 8(8%), *Salmonella* HilA homolog (*eilA*) 7(7%), aggregative adherence regulator (*aggR*) 5(5%), anti-aggregation protein transporter (*aatA*) 4(4%), AAF/I fimbrial subunit (*aggA*) 3(3%) and AAF/IV fimbrial subunit (*agg4a*) 2(2%). Genes encoding the commonly encountered AAF/II, AAF/III and AAF/V fimbriae were not found in any hybrid isolate.

Four of the isolates represented EPEC- enterotoxigenic *E. coli* (ETEC) hybrids, carrying heat labile endotoxin virulence genes (*ltcA*), one of two toxin genes that define ETEC, were identified and one of these was an EPEC-EAEC-ETEC hybrid that also carried *eilA*, air and *capU* genes (Fig 1). Five isolates carried *afa* genes. Three of these isolates were EPEC-EAEC hybrids and one of them was the EPEC-EAEC-ETEC hybrid (Fig 3).

Altogether, 18 (18.8%) isolates in the study carried genes that define other diarrheagenic *E. coli* pathotypes, in addition to the EPEC virulence genes. Only one of these hybrid strains was a typical EPEC isolate, positive for EAF plasmid genes *bfp* and *per*. Hybrid genotypes were distributed across the phylogeny (five, five, five, one and two, respectively belonged to *E. coli* phylogroups A, B1, D, E and untypeable). Nine of the eighteen hybrid isolates were recovered from children with diarrhea.

## EPEC outbreaks

Multiple EPEC clades comprised genomes separated with 20 or less SNPs. ST28 EPEC isolates (O119:H6) were recovered from three pediatric patients presenting at the same referral hospital. Another cluster of five atypical EPEC serogroup O127:H29 ST7798 isolates were from patients, all aged 3 months, attending primary health care centers (n = 2) and the

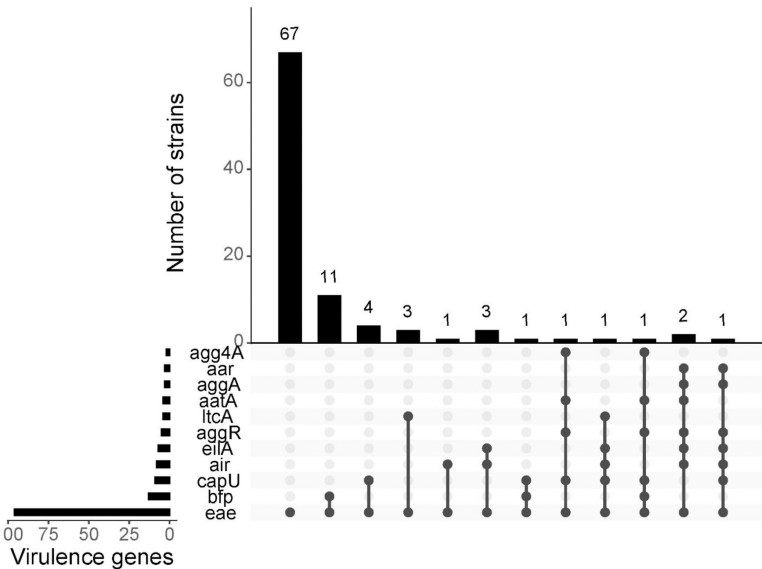

**Fig 3. EPEC hybrids carrying pAA genes that define the EAEC pathotype (*air, eilA, aggR, aap, aggA, agg4A, aar* and *capU*) and/or ETEC (*ltcA*)-defining virulence genes.**

referral hospital (n = 3) in Ibadan. Two of the children at the referral center presented with diarrhea in 2017 and one was a healthy control. These isolates differed by only 2 SNPs and the time and place data suggest that they are epidemiologically connected and so represent a localized outbreak.

ST 517 EPEC was the most prevalent (n = 25, 26%) ST from this study and they were all atypical (Fig 1). As shown in Fig 2, ST517 EPEC was isolated from both cases and control children's specimens from IFE, PHC and HIV patients in IDI but none from UCH. All ST517 isolates carried *eae* epsilon genes. There are three sub-clades within the ST517 clade in this study. The first is comprised of O171:H19 isolates from healthy children attending primary health centers in Ibadan and is related to the second clade of O165:H9 and O109:H9 isolates from children or PLWHIV attending clinics in Ibadan. The third and largest clade is comprised of O171:H19 isolates from children recruited in Ife. Four of these were from children with diarrhea and 11 were from controls (Fig 4). When the number of SNP distances among these isolates was computed, it was observed that the ST517 O171:H19 EPEC subclade from children in Ife is made up of isolates <10 SNPs apart and they were all recovered from specimens obtained between the 4th of August and the 21st of September 2017 (Fig 4a). Three of the four case isolates and six of the eleven isolates from asymptomatic individuals were all recovered from specimens collected on the 28th of August 2017 and all of them resided within a circle with a 1.2 km radius.

All the ST517 likely outbreak isolates from Ile-Ife had, in addition to a LEE encoding intimin epsilon, the *sepA, senB, toxB, espC, nleB* and *nleC* virulence genes. They also harbored genes conferring resistance to extended-spectrum β-lactam drugs, sulfonamide and tetracycline (Fig 4b). Notably, no virulence loci were more predominant in case isolates, compared to controls (p > 0.05). Many children were infected in this outbreak but the numbers were still too low to make firm conclusions about associations, particularly among subgroups. Nonetheless, we were able to make some interesting observations in the context of the outbreak. The children from whom likely outbreak strains were recovered that had diarrhea were aged 8, 16, 18 and 24 months. Three of the children without diarrhea were aged 12 months and the rest were aged 24–36 months (Fig 4c). While undernutrition was common as a whole, proportionately more of the children who were not stunted carried ST517 EPEC outbreak isolates without symptoms. Two children who succumbed to diarrhea showed lower weights for age and heights for age than healthy children stunted growth (Fig 4d).

## Discussion

Sequence-based characterization of EPEC from Africa is uncommon and therefore little is known about circulating lineages, particularly after classical group serotyping, which had notably low sensitivity in Africa, and was no longer advocated for routine use [54]. In this study, which identified and subtyped EPEC by whole genome sequencing, typical EPEC, and EPEC belonging to classical serotypes, were uncommon. Majority of the EPEC isolates examined in this study 67 (69.8%) belong to newly identified phylogenomic lineages, in keeping with reports of EPEC from recent Africa and Asia studies [8,50,55]. Our findings highlight the diversity of the EPEC pathogroup, reflect its convergent origins and continuous evolution, and demonstrate that more studies need to be conducted in the parts of the world most burdened by EPEC disease [8,35,50,55,56].

In this study, as in many other recent African studies [57], the EPEC isolates detected from Ibadan and Ife studies were mostly atypical EPEC 83(86.5%), which are emerging enteropathogens with global distribution [58–60]. Several countries have experienced EPEC outbreaks caused by atypical EPEC [29–31,61] in recent years leading some to suspect that they are replacing typical EPEC as a leading cause of diarrhea in both developing and industrialized nations [58–60]. ([29–31,61]. The number of reports in the last decade indexed in Medline is at least three times that for the decade before with a greater diversity of aEPEC, including hybrid strains, being identified in outbreaks in recent years [62,63]. It must however be conceded that at least some of the increased reporting arises from development and deployment of molecular tools that can detect aEPEC that would earlier have been missed. The historical epidemiology of aEPEC, particularly in Africa, is difficult to understand as earlier studies used diagnostic methods biased towards typical EPEC. Typical EPEC

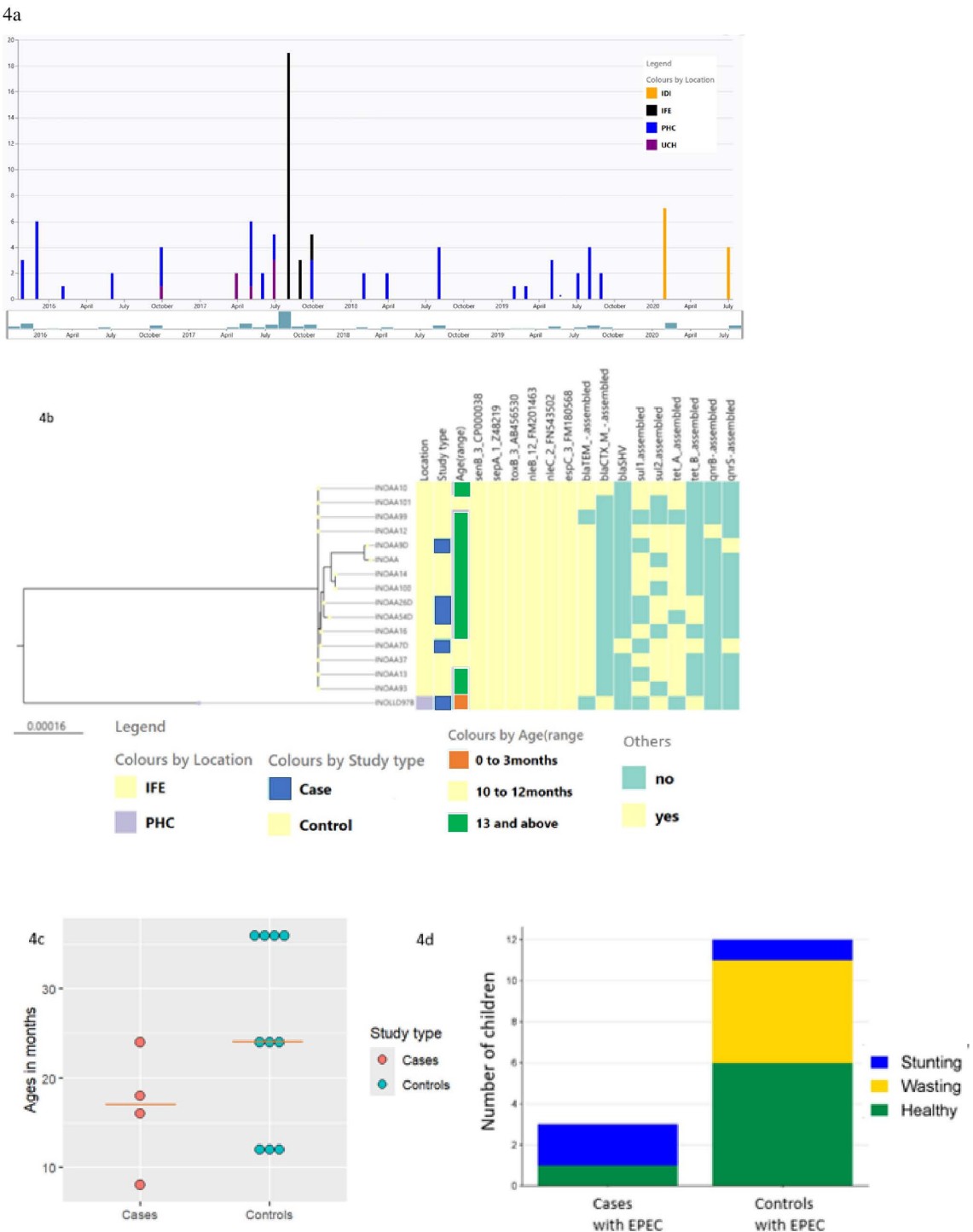

**Fig 4. Characterization of a cluster of ST517 EPEC recovered in Ile-Ife. (a).** Epicurve showing the dates of enrollment of children with diarrhea (top) and those from whom ST517 isolates were recovered (bottom). **(b).** Virulence and antimicrobial resistance genes in ST517 isolate from Ife that, based on sharing <10 SNPs, are believed to constitute an outbreak. **(c).** Age distribution of individuals from whom EPEC included in the tree in (b) were recovered with the red line representing the median age for each category. **(d).** The nutritional status of children from whom isolates in (b) above were collected.

were uncommonly recovered in this study, but were more commonly isolated from children with diarrhea (p < 0.05) and four of the seven affected children were enrolled at referral centres, suggesting disease severity.

While most classical EPEC serogroups are typical EPEC strains, they can also include atypical EPEC and other DEC pathotypes, such as EAEC [26,16,64]. Specific O:H types are more reliable markers of EPEC than O-groups alone and often represent lineages disseminated globally particularly classical strains [15,18,21]. Molecular tools are now more commonly used for classification, but certain O:H serotypes remain closely associated with EPEC and have been useful in distinguishing them from other DEC pathotypes. In this study, we recovered O119:H6 and O142:H34 isolates, which represent well-described serotypes of typical EPEC [17,65,66] as well as two EPEC-EAEC hybrids belonging to O86:H18 a serovar originally described as EPEC and then re-classified as EAEC [67]. A number of isolates belonged to O-groups commonly associated with EHEC, including O55, O157 and O118 were isolated but none of these isolates had Shiga-toxin genes, adding to the evidence that sub-classifying DEC by serotyping in our setting could lead to pathotype miscalls [68,69]. The majority, 88 (91.7%), of the identified EPEC in this study were non-classical EPEC.

It is challenging to find historical context of the data gathered in our study because previous studies in Nigeria lack whole genome-sequence data for EPEC or indeed archived the isolates. Serotyping was the most commonly employed delineation tool. Unless performed at a reference centre, serotyping can be inaccurate and only finds strains belonging to classical groups or types [2]. We can glean some learnings when we juxtapose our data on data from studies that used robust serotyping methods. Ifeanyi et al. (2017) identified many EPEC serovarieties and recovered many isolates they deemed non-typeable, suggested that they might belong to non-familiar serotypes. While rarely reported elsewhere, O33:H6/H34 EPEC were found at low frequency in all studies [31,32], including this one (belonging to ST8747), indicating the possibility that they represent a local endemic clade.

The most prevalent intimin subtypes among aEPEC isolates of various serotypes worldwide have been reported to be Alpha (α), Beta (β), Gamma (γ), Zeta (ζ), Delta (δ) and Epsilon (ε), while other intimin subtypes are said to be less common [5,25,70]. In this study all the stated intimin types were identified except for Delta and Zeta. Moreover, 38 (39.6%) of the isolates carried Iota2, Kappa, Lambda, Rho, Theta and Theta2 subtypes. An ST378 phylogroup isolate is the only one bearing the Iota allele of intimin and is related to a recently described atypical EPEC10 clone from Brazil shown to induce mucus hypersecretion [53]. The occurrence of different intimin subtypes confirms the diversity of EPEC isolates circulating in Nigeria and their under-representation among well-studied strains. We have previously reported that consensus intimin PCR primers target Iota, Rho, and Theta alleles with low efficiency, and their abundance in our strain set emphasizes the need to develop diagnostic tools that are inclusive of or specific for prominent West African lineages. We note that the intimin Iota2-bearing ST378 isolate we identified is closely related to a hypervirulent clone reported from Gambia [9] and therefore these lineage remains relatively under-characterized in the literature, with limited genomic or functional data available.

The number and variety of hybrid isolates detected in this study demonstrates the complex evolutionary dynamics of EPEC, and more generally, to the *E. coli* species. Hybrid EPEC were more common than, and did include, EPEC isolates carrying the EPEC adherence plasmid that defined typical EPEC. In this regard, EPEC carrying virulence genes other than *bfp* have greater epidemiological significance in our setting. This may be true in other parts of the world as hybrid strains have been commonly reported recently and are often associated with outbreaks or illness severity [62,71,72]. In this study, they were found throughout the phylogeny and therefore evolution by independent horizontal acquisition of relevant virulence genes may be common. Nine of 18 hybrid isolates were from children with diarrhea, and therefore hybrids (and how they evolve) should be the subject of future research.

We identified two clonal clusters, ST28 (O119:H6, typical EPEC) and an ST7798 (O127:H29, atypical EPEC), appear to be connected to outbreaks, both including children presenting at a referral hospital in Ibadan, as well as an ST517 outbreak that was picked up from children attending an Ile-Ife primary health care center. Interestingly two apparently healthy children recruited from an Ibadan primary health care centre in 2015 yielded very similar ST7798 isolates, being

only 10–11 SNPs away from the hospital isolates suggesting that this ST7798 atypical lineage may be circulating more broadly in the community across Ibadan and that confirming EPEC outbreaks with genomic information alone may not be possible. All three likely outbreaks revealed that healthy carriers as well as ill children were colonized. The most prominent lineage in this study was an ST517 EPEC clade of closely related isolates that was shown to represent the largest outbreak that occurred in Ile-Ife in August/September 2017. ST517 strains are not commonly reported in the literature and this clade has yet to be assigned an EPEC phylogroup. One report, from China, implicated imported foods [73] and O71:H19 EPEC has been identified from human stool samples in Brazil, South Korea and in dog feces from Minnesota [74–76], implying that O71:H19 ST571 EPEC are broadly disseminated and may colonize and/or cause disease humans and animals. As many as 17 children cultured in that study carried the strain but only four were diarrhea patients [38]. It was observed that there were no easily apparent differences among the bacterial isolates from children with diarrhea and asymptomatic children in this study, however, those children with diarrhea were stunted in growth. However, we cannot rule out the potential contribution of other bacterial, viral, or parasitic infections in these children. Altogether, our data suggest that EPEC outbreaks can occur undetected and that when they do, the most vulnerable are at highest risk.

## Conclusion

EPEC isolates circulating in Ile-Ife and Ibadan, Nigeria are diverse and include globally disseminated clones of known virulence as well as EPEC that differ substantially from well-characterized lineages reported from other parts of the world. EPEC isolates that carry EAEC, ETEC or DAEC genes are common in our study and the current definitions that classify all EAF/*bfp*-negative strains as 'atypical' support the idea that very little is known about the pathogenesis of EPEC lineages, which are endemic in southwestern Nigeria. Genomic surveillance can uncover EPEC outbreaks, the scale and frequency of which are largely unknown and grossly under-reported. As infected people may be asymptomatic and our data suggest that vulnerable individuals such as nutritionally deficient children may bear the brunt of EPEC disease, there is need for active, continued genomic surveillance of the epidemiology of EPEC in our setting.

## Supporting information

**S1 Table. Demographic information about patients from which EPEC in this study were derived and characteristics derived from genomes including sequence types, serovars, phylogenetic groups, virulence loci and antimicrobial resistance genes.**
(CSV)

**S2 Table. Source and properties of locally-derived and international EPEC genomes.**
(XLSX)

## Acknowledgments

We thank Stella Ekpo, Amos Olowookere, Justice C Onwuka, Mariam Odebode, Elizabeth Akande and El-shama Q Nwoko for technical assistance. We are grateful to clinicians, notably Drs A Adepoju, BO Ogunbosi, KO Akande and T Ilori, and consenting patients who enabled the epidemiological studies from which the isolates were obtained.

## Author contributions

**Conceptualization:** Aaron O Aboderin, Iruka N Okeke.

**Data curation:** Olabisi C Akinlabi, Rotimi A Dada, Ademola A Olayinka, Iruka N Okeke.

**Formal analysis:** Olabisi C Akinlabi, Rotimi A Dada, Ademola A Olayinka, Ibukunoluwa O Oginni-Falajiki, Oyeniyi S Bejide, Iruka N Okeke.

**Funding acquisition:** Nicholas R Thomson, Iruka N Okeke.

**Investigation:** Olabisi C Akinlabi, Rotimi A Dada, Ademola A Olayinka, Ibukunoluwa O Oginni-Falajiki, Oyeniyi S Bejide, Pelumi D Adewole, Iruka N Okeke.

**Methodology:** Olabisi C Akinlabi, Rotimi A Dada, Pelumi D Adewole, Aaron O Aboderin, Iruka N Okeke.

**Project administration:** Olabisi C Akinlabi, Rotimi A Dada, Ademola A Olayinka, Aaron O Aboderin, Iruka N Okeke.

**Resources:** Nicholas R Thomson, Iruka N Okeke.

**Software:** Rotimi A Dada, Nicholas R Thomson.

**Supervision:** Nicholas R Thomson, Aaron O Aboderin, Iruka N Okeke.

**Validation:** Olabisi C Akinlabi, Rotimi A Dada, Ademola A Olayinka, Iruka N Okeke.

**Visualization:** Olabisi C Akinlabi, Rotimi A Dada, Iruka N Okeke.

**Writing – original draft:** Olabisi C Akinlabi, Iruka N Okeke.

**Writing – review & editing:** Olabisi C Akinlabi, Rotimi A Dada, Ademola A Olayinka, Ibukunoluwa O Oginni-Falajiki, Oyeniyi S Bejide, Pelumi D Adewole, Nicholas R Thomson, Aaron O Aboderin, Iruka N Okeke.

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
