## [Decision Letter · Decision Letter 0]

28 Mar 2025

PNTD-D-25-00292

Genomic epidemiology of enteropathogenic Escherichia coli (EPEC) in southwestern Nigeria

Dear Dr. Okeke,

Thank you for submitting your manuscript to PLOS Neglected Tropical Diseases. After careful consideration, we feel that it has merit but does not fully meet PLOS Neglected Tropical Diseases's publication criteria as it currently stands. Therefore, we invite you to submit a revised version of the manuscript that addresses the points raised during the review process.

Please submit your revised manuscript within 60 days. If you will need more time than this to complete your revisions, please reply to this message or contact the journal office at plosntds@plos.org. Please include the following items when submitting your revised manuscript:

We look forward to receiving your revised manuscript.

Kind regards,

Kirkby D Tickell, MBBS BSc

Academic Editor

Ana LTO Nascimento

Section Editor

Shaden Kamhawi

co-Editor-in-Chief

Paul Brindley

co-Editor-in-Chief

**Journal Requirements:**

At this stage, the following Authors/Authors require contributions: Oyeniyi S Bejide, and Nicholas R Thomson. Please ensure that the full contributions of each author are acknowledged in the "Add/Edit/Remove Authors" section of our submission form.

**Reviewers' Comments:**

Reviewer's Responses to Questions

**Key Review Criteria Required for Acceptance?**

**Methods:**

-Are the objectives of the study clearly articulated with a clear testable hypothesis stated?

-Is the study design appropriate to address the stated objectives?

-Is the population clearly described and appropriate for the hypothesis being tested?

-Is the sample size sufficient to ensure adequate power to address the hypothesis being tested?

-Were correct statistical analysis used to support conclusions?

-Are there concerns about ethical or regulatory requirements being met?

Reviewer #1: See below

Reviewer #2: There was a considerable lack of detail in the methods - absence of EPEC identification methods/ inclusion criteria, lack of whole genome sequencing methods - platform (MiSeq ? NextSeq?, library prep, etc), lack of details around genome assembly and tools used for genetic characterisation (more detail in attached file).

No criteria/method for eae allele typing or phylotyping provided.

No statistical methods provided.

No phylogenetic tree construction methods provided.

Not stated where the international strains were sourced from, nor any accession numbers provided.

No supplementary file/table provided with individual isolate metadata and Biosample numbers.

**Results:**

-Does the analysis presented match the analysis plan?

-Are the results clearly and completely presented?

-Are the figures (Tables, Images) of sufficient quality for clarity?

Reviewer #1: See below

Reviewer #2: The majority of of genomic characterisation and analyses in this manuscript are well executed, I have concerns regarding some errors in the manuscript (discrepancies in serotyping, interpretation of E coli pathotypes based on carriage of particular virulence genes, etc).

Figure quality could be improved.

See attached file.

**Conclusions:**

-Are the conclusions supported by the data presented?

-Are the limitations of analysis clearly described?

-Do the authors discuss how these data can be helpful to advance our understanding of the topic under study?

-Is public health relevance addressed?

Reviewer #1: See below

Reviewer #2: The study broad study conclusions are acceptable - The study determined that both classical and non-classical lineages of EPEC are circulating and the predominant pathotype is aEPEC, rather than tEPEC. This follows patterns seen elsewhere, such as in Brazil.

However, the conclusions regarding hybrid pathotypes contains errors, as they are based on flawed interpretation of virulence gene carriage.

The discussion also included information that was repetitive of the introduction, and additional background which should be shifted to the introduction section.

**Editorial and Data Presentation Modifications?**

Reviewer #1: See below

Reviewer #2: (No Response)

**Summary and General Comments:**

Reviewer #1: PNTD-D-25-00292

"Genomic epidemiology of enteropathogenic Escherichia coli (EPEC) in southwestern Nigeria."

Akinlabi, A.C., et al.

This is an interesting manuscript about characterization of the circulating enteropathogenic Escherichia coli (EPEC) strains in southwestern Nigeria. In addition to previously reported typical EPEC and atypical EPEC isolates, the study identified novel serotypes and association with potential outbreaks identified. The authors should strongly consider the following points to improve the paper:

Major points:

1. It might be important to clarify the differences between classical and non-classical EPEC vs typical and atypical EPEC (e.g. table 3). For the not expert reader, this is confusing. Do classical EPEC include tEPEC and aEPEC? How about non-classical?

2. Introduction and discussion, it might be important to discuss the trend happening worldwide of more aEPEC casing more outbreaks and cases than tEPEC. Is Africa seeing the same trend as in Europe? Such comparisons have been made in Clin Microb Infect, 2015. 21:729-734.

3. Lanes 135-138, results (lanes 304-307) and Fig 4D, unfortunately the sample size is so small that conclusions associated with stunting and wasting are not fully supported. It is well known from the GEM study that diarrheal infectious by more than one pathogen, not just EPEC, are associated with stunting and wasting and in the current study they cannot rule out the presence of a parasitic or viral infection or other bacteria in these patients. Please revise.

4. Results and Fig 1, are intimin alleles in correlation to what is described as EPEC intimin alleles? It is not evident in the results section. For example, in lane 275, what is the relevance of identifying intimin epsilon? For example, lanes 368-370 should be incorporated in the introduction or results to add context.

5. Fig 1 and lanes 244-246, is the presence of one gene (ltcA) sufficient to define the strain as EPEC-ETEC hybrid strain?

6. Lanes 202-203 and Fig 2, the closely relatedness to Gambian samples is not evident from the data as presented.

Minor points:

1. Abstract, lane 38, “…were the most prevalent…”

2. Lane 68, do you mean “pathogroups” instead of “subcategories”

3. Lane 75 “The serovarieties represent….

4. Lane 188 and fig 1, the serotype O142:H34 is missing from Fig. 1 list of serotypes.

5. Lane 208 “isolation sources and location…”

6. Fig. 2 the external ring, at the bottom of Fig. 2 is not fully visible.

7. Color code in fig 4B is confusing because it uses green and yellow in most conditions tested. Perhaps study type and age range could be depicted with a different color.

8. Lanes 310-311, “…with isolates of ETEC, EAEC and EPEC being…”

9. Lane 315, what do you mean with “insensitive”? do you mean “absent”?

10. Lane 320, “…of the EPEC pathogroup…”

11. Lane 340, “…12 EPEC serogroups…”

12. Lane 379 “unfamiliar to science”? Since the Gambian isolated is reported, this might not be considered unfamiliar to science.

13. Lane 382, “…EPEC, and more generally, to the Ecoli…”

Reviewer #2: This study investigates the diversity and characteristics of EPEC in Nigeria. Given that diarrhoeal disease caused by EPEC is still a health problem in infants in many places including Africa, it is important to understand the evolution of these pathogens.

Overall, I cannot recommend this manuscript for publication as it is currently written. Multiple major corrections are required, data errors need to be checked/corrected, and additional supplementary data needs to be provided.

PLOS authors have the option to publish the peer review history of their article (what does this mean? ). If published, this will include your full peer review and any attached files.

**Do you want your identity to be public for this peer review?** For information about this choice, including consent withdrawal, please see our Privacy Policy .

Reviewer #1: No

Reviewer #2: No

**Figure resubmission:**
---

## [Decision Letter · Decision Letter 1]

26 Jun 2025

PNTD-D-25-00292R1

Genomic epidemiology of enteropathogenic Escherichia coli in southwestern Nigeria

Dear Dr. Okeke,

Thank you for submitting your manuscript to PLOS Neglected Tropical Diseases. After careful consideration, we feel that it has merit but does not fully meet PLOS Neglected Tropical Diseases's publication criteria as it currently stands. Therefore, we invite you to submit a revised version of the manuscript that addresses the points raised during the review process.

Please note the additional comment below and submit your revised manuscript within 60 days . If you will need more time than this to complete your revisions, please reply to this message or contact the journal office at plosntds@plos.org. Please include the following items when submitting your revised manuscript:

We look forward to receiving your revised manuscript.

Kind regards,

Kirkby D Tickell, MBBS BSc

Academic Editor

Ana LTO Nascimento

Section Editor

Shaden Kamhawi

co-Editor-in-Chief

Paul Brindley

co-Editor-in-Chief

**Additional Editor Comments :**

Please note that one of the Reviewer comments do not appear to have been addressed in the response. Apologies if this is an error within our system to either communicate these comments, or display the revisions you have made. The comments should be attached, but I will also copy them here:

I do not understand the reason the authors did not respond to my prior comments but without a point-by-point responses to my queries, I cannot evaluate the manuscript. here are the prior comments for their reference.

This is an interesting manuscript about characterization of the circulating enteropathogenic Escherichia coli (EPEC) strains in southwestern Nigeria. In addition to previously reported typical EPEC and atypical EPEC isolates, the study identified novel serotypes and association with potential outbreaks identified. The authors should strongly consider the following points to improve the paper:

Major points:

1. It might be important to clarify the differences between classical and non-classical EPEC vs typical and atypical EPEC (e.g. table 3). For the not expert reader, this is confusing. Do classical EPEC include tEPEC and aEPEC? How about non-classical?

2. Introduction and discussion, it might be important to discuss the trend happening worldwide of more aEPEC casing more outbreaks and cases than tEPEC. Is Africa seeing the same trend as in Europe? Such comparisons have been made in Clin Microb Infect, 2015. 21:729-734.

3. Lanes 135-138, results (lanes 304-307) and Fig 4D, unfortunately the sample size is so small that conclusions associated with stunting and wasting are not fully supported. It is well known from the GEM study that diarrheal infectious by more than one pathogen, not just EPEC, are associated with stunting and wasting and in the current study they cannot rule out the presence of a parasitic or viral infection or other bacteria in these patients. Please revise.

4. Results and Fig 1, are intimin alleles in correlation to what is described as EPEC intimin alleles? It is not evident in the results section. For example, in lane 275, what is the relevance of identifying intimin epsilon? For example, lanes 368-370 should be incorporated in the introduction or results to add context.

5. Fig 1 and lanes 244-246, is the presence of one gene (ltcA) sufficient to define the strain as EPEC-ETEC hybrid strain?

6. Lanes 202-203 and Fig 2, the closely relatedness to Gambian samples is not evident from the data as presented.

Minor points:

1. Abstract, lane 38, “…were the most prevalent…”

2. Lane 68, do you mean “pathogroups” instead of “subcategories”

3. Lane 75 “The serovarieties represent….

4. Lane 188 and fig 1, the serotype O142:H34 is missing from Fig. 1 list of serotypes.

5. Lane 208 “isolation sources and location…”

6. Fig. 2 the external ring, at the bottom of Fig. 2 is not fully visible.

7. Color code in fig 4B is confusing because it uses green and yellow in most conditions tested. Perhaps study type and age range could be depicted with a different color.

8. Lanes 310-311, “…with isolates of ETEC, EAEC and EPEC being…”

9. Lane 315, what do you mean with “insensitive”? do you mean “absent”?

10. Lane 320, “…of the EPEC pathogroup…”

11. Lane 340, “…12 EPEC serogroups…”

12. Lane 379 “unfamiliar to science”? Since the Gambian isolated is reported, this might not be considered unfamiliar to science.

13. Lane 382, “…EPEC, and more generally, to the Ecoli…”

**Journal Requirements:**

1) We noticed that you used the phrase 'unpublished ' in the manuscript. We do not allow these references, as the PLOS data access policy requires that all data be either published with the manuscript or made available in a publicly accessible database. Please amend the supplementary material to include the referenced data or remove the references.

2) Your current Financial Disclosure includes "UK Medical Research Council (MRC) and the UK Department for International Development (DFID) under the MRC/DFID 23 concordat agreement that is also part of the EDCTP2 programme supported by the European Union." If you received funding from the aforementioned sources, please include it in the Funding Information tab. Please also ensure to include the initials of the author(s) who received the fund(s). 

NOTE: Please ensure that the funders and grant numbers match between the Financial Disclosure field and the Funding Information tab in your submission form. Note that the funders must be provided in the same order in both places as well.

3) Thank you for indicating that "The authors declare no competing interests." Please state "The authors have declared that no competing interests exist". 

**Comments to the Authors:**

**Please note that one of the reviews is uploaded as an attachment.**

**Reviewers' Comments:**

Reviewer's Responses to Questions

**Key Review Criteria Required for Acceptance?**

**Methods**

-Are the objectives of the study clearly articulated with a clear testable hypothesis stated?

-Is the study design appropriate to address the stated objectives?

-Is the population clearly described and appropriate for the hypothesis being tested?

-Is the sample size sufficient to ensure adequate power to address the hypothesis being tested?

-Were correct statistical analysis used to support conclusions?

-Are there concerns about ethical or regulatory requirements being met?

Reviewer #1: see below

Reviewer #2: (No Response)

**Results**

-Does the analysis presented match the analysis plan?

-Are the results clearly and completely presented?

-Are the figures (Tables, Images) of sufficient quality for clarity?

Reviewer #1: see below

Reviewer #2: (No Response)

**Conclusions**

-Are the conclusions supported by the data presented?

-Are the limitations of analysis clearly described?

-Do the authors discuss how these data can be helpful to advance our understanding of the topic under study?

-Is public health relevance addressed?

Reviewer #1: see below

Reviewer #2: (No Response)

**Editorial and Data Presentation Modifications?**

Reviewer #1: see below

Reviewer #2: (No Response)

**Summary and General Comments**

Reviewer #1: I do not understand the reason the authors did not respond to my prior comments but without a point-by-point responses to my queries, I cannot evaluate the manuscript. here are the prior comments for their reference.

This is an interesting manuscript about characterization of the circulating enteropathogenic Escherichia coli (EPEC) strains in southwestern Nigeria. In addition to previously reported typical EPEC and atypical EPEC isolates, the study identified novel serotypes and association with potential outbreaks identified. The authors should strongly consider the following points to improve the paper:

Major points:

1. It might be important to clarify the differences between classical and non-classical EPEC vs typical and atypical EPEC (e.g. table 3). For the not expert reader, this is confusing. Do classical EPEC include tEPEC and aEPEC? How about non-classical?

2. Introduction and discussion, it might be important to discuss the trend happening worldwide of more aEPEC casing more outbreaks and cases than tEPEC. Is Africa seeing the same trend as in Europe? Such comparisons have been made in Clin Microb Infect, 2015. 21:729-734.

3. Lanes 135-138, results (lanes 304-307) and Fig 4D, unfortunately the sample size is so small that conclusions associated with stunting and wasting are not fully supported. It is well known from the GEM study that diarrheal infectious by more than one pathogen, not just EPEC, are associated with stunting and wasting and in the current study they cannot rule out the presence of a parasitic or viral infection or other bacteria in these patients. Please revise.

4. Results and Fig 1, are intimin alleles in correlation to what is described as EPEC intimin alleles? It is not evident in the results section. For example, in lane 275, what is the relevance of identifying intimin epsilon? For example, lanes 368-370 should be incorporated in the introduction or results to add context.

5. Fig 1 and lanes 244-246, is the presence of one gene (ltcA) sufficient to define the strain as EPEC-ETEC hybrid strain?

6. Lanes 202-203 and Fig 2, the closely relatedness to Gambian samples is not evident from the data as presented.

Minor points:

1. Abstract, lane 38, “…were the most prevalent…”

2. Lane 68, do you mean “pathogroups” instead of “subcategories”

3. Lane 75 “The serovarieties represent….

4. Lane 188 and fig 1, the serotype O142:H34 is missing from Fig. 1 list of serotypes.

5. Lane 208 “isolation sources and location…”

6. Fig. 2 the external ring, at the bottom of Fig. 2 is not fully visible.

7. Color code in fig 4B is confusing because it uses green and yellow in most conditions tested. Perhaps study type and age range could be depicted with a different color.

8. Lanes 310-311, “…with isolates of ETEC, EAEC and EPEC being…”

9. Lane 315, what do you mean with “insensitive”? do you mean “absent”?

10. Lane 320, “…of the EPEC pathogroup…”

11. Lane 340, “…12 EPEC serogroups…”

12. Lane 379 “unfamiliar to science”? Since the Gambian isolated is reported, this might not be considered unfamiliar to science.

13. Lane 382, “…EPEC, and more generally, to the Ecoli…”

Reviewer #2: The revised manuscript is much improved in content and, structure and flows well now. Thank you for providing the additional methods information, references and supplementary data files.

This is an important study in a very under-explored area of diarrhoeal disease in human infants in Nigeria and has global implications.

However, there are still multiple minor errors (generally grammatical) and clarifications required in some sections. Also, please correct the typical EPEC STs listed in the results.

Recommend accept with minor revision.

Please see attached file for specific manuscript comments.

PLOS authors have the option to publish the peer review history of their article (what does this mean? ). If published, this will include your full peer review and any attached files.

**Do you want your identity to be public for this peer review?** For information about this choice, including consent withdrawal, please see our Privacy Policy .

Reviewer #1: No

Reviewer #2: No

**Figure resubmission:**
---

## [Editor Report · Decision Letter 2]

4 Aug 2025

Dear Dr. Okeke,

We are pleased to inform you that your manuscript 'Genomic epidemiology of enteropathogenic Escherichia coli in southwestern Nigeria' has been provisionally accepted for publication in PLOS Neglected Tropical Diseases.

Best regards,

Kirkby D Tickell, MBBS BSc

Academic Editor

Ana LTO Nascimento

Section Editor

Shaden Kamhawi

co-Editor-in-Chief

Paul Brindley

co-Editor-in-Chief

---

## [Editor Report · Acceptance letter]

Dear Prof Okeke,

We are delighted to inform you that your manuscript, " 

Genomic epidemiology of enteropathogenic *Escherichia coli*  in southwestern Nigeria," has been formally accepted for publication in PLOS Neglected Tropical Diseases.

Best regards,

Shaden Kamhawi

co-Editor-in-Chief

Paul Brindley

co-Editor-in-Chief
